# The Health Needs of Regionally Based Individuals Who Experience Homelessness: Perspectives of Service Providers

**DOI:** 10.3390/ijerph19148368

**Published:** 2022-07-08

**Authors:** Grace Bennett-Daly, Hazel Maxwell, Heather Bridgman

**Affiliations:** 1School of Nursing, University of Tasmania, Newnham, Launceston, TAS 7248, Australia; grace.bennettdaly@utas.edu.au; 2School of Health Sciences, University of Tasmania, Rozelle, Sydney, NSW 2015, Australia; 3Centre for Rural Health, University of Tasmania, Newnham, Launceston, TAS 7248, Australia; heather.bridgman@utas.edu.au

**Keywords:** homelessness, health care, nursing, nurse-led

## Abstract

The bidirectional relationship between homelessness and poor health and the barriers that individuals who experience homelessness face when trying to access healthcare are well documented. There is, however, little Australian research exploring the situation of individuals who experience homelessness in regional contexts and, moreover, from the perspective of service providers. A qualitative descriptive methodology underpinned this study, with in-depth semi-structured interviews being conducted with 11 service providers to identify barriers to care faced by people who experience homelessness and barriers that service providers themselves experience in supporting this population. The key barriers identified were client-level barriers: living day-by-day, financial, health literacy, mental health conditions, behaviour, safety and stigma; provider-level barriers: few bulk-billing doctors, fragmented services, limited resources, negative past experiences with healthcare; and system level barriers: transportation, over-stretched healthcare services. The combined impact of these barriers has significantly contributed to the desperate situation of people experiencing homelessness in Launceston. This situation is likely replicated in other regional populations in Australia. Given that individuals experiencing homelessness have higher rates of every measure in health inequality, steps need to be taken to reduce barriers, and a standardised approach to health care urgently needs to be implemented by governments at the state and national level to improve the health of regionally based individuals experiencing homelessness.

## 1. Introduction

Individuals experiencing homelessness and those at risk of becoming homeless are among the most socially and economically disadvantaged populations worldwide [1,2,3]. This precarity negatively affects an individual’s physical and mental health, as well as their education and employment opportunities [4]. One of the key complicating factors in understanding homelessness in regional Australia is that variation exists within the different national estimates of homelessness due to the absence of an agreed definition of homelessness. In addition, there is the difficulty of obtaining accurate data due to both the stigma attached to homelessness and the nomadic existence of this population [5]. The current working definition used by most agencies in Australia is that an individual is classified as *homeless*:

*“if their current living arrangement is in a dwelling that is inadequate; or has no tenure, or if their initial tenure is short and not extendable; or does not allow them to have control of, and access to space for social relations”*.[6]

Quantifying the extent of the homelessness issue in Australia is therefore challenging. The most recently available data from the 2016 census estimate that more than 116,000 individuals are homeless [7]. This represents an increase of more than 13% since the 2011 census [8]. Whilst there are no official statistics related to the number of individuals experiencing homelessness since COVID, it is believed that due to increased unemployment, underemployment, lack of housing infrastructure and rising rental costs, more individuals have since become homeless or are in real danger of becoming homeless [9]. Although there have not been any reports outlining the rates of homelessness since the COVID-19 pandemic began, it has been predicted that there has been a considerable spike now that the government coronavirus financial support schemes and housing protection initiatives have been withdrawn [9,10]. Of note, use of family and domestic violence services has increased significantly since the beginning of the COVID-19 pandemic [11,12]. Family and domestic violence is the leading cause of homelessness for women and children in Australia [13]. Compounding this issue is that there was also a reported reduced opportunity for affected women and children to engage with family and domestic violence services due to the implementation of lockdowns [14]. In addition, older women are believed to be at significant risk of homelessness due to reduced employment opportunities following the COVID-19 pandemic [15]. For all of these reasons, homelessness has come into sharp focus, giving added impetus and urgency for the Australian government to address the housing and healthcare issues of individuals and families experiencing homelessness [9,10].

The bidirectionality of the relationship between homelessness and poor health is complex [16,17,18]. Some individuals become homeless as a result of their health condition(s); for others, homelessness can also lead to and exacerbate mental and physical health conditions [19]. This compounding causal reciprocity has been observed worldwide [19,20,21,22]. Regardless of which issue preceded the other, once an individual becomes homeless, managing their health condition(s) becomes unviable, and the likelihood of developing additional health problems increases dramatically [21,22,23,24]. Individuals experiencing homelessness have significantly higher rates of nearly any measure of health inequality [16,20,21,25,26,27], with reported prevalence rates of mental health problems ranging from 4% to 100% [28,29,30,31,32,33]. Individuals experiencing homelessness also have disproportionally higher rates of poor physical health compared with the general population, with reported prevalence rates ranging from 41% to 76% [29,34,35,36,37].

A consistent finding in the literature is that despite having a clear need for ongoing healthcare support, individuals experiencing homelessness are often not registered with a general practitioner (GP) nor are linked to primary and preventative healthcare services [38]. This non-engagement with services often means that individuals experiencing homelessness become more seriously ill and are more likely to experience premature death than the general population [39,40]. Reduced utilisation of healthcare services can largely be attributed to often insurmountable barriers, which this population faces across all levels [41], including challenges related to access, experiencing stigma and discrimination [42], financial constraints, inadequate transportation [43,44,45] and low health literacy [46]. The issue of access to free or affordable health care repeatedly arises as a key barrier for individuals experiencing homelessness [19,47]. In Australia, medical expenses can be partially or completely covered under a national healthcare insurance scheme (Medicare) involving a bulk billing scheme where healthcare providers directly bill the government for their services rather than charge patients [48]. Despite all health care providers being able to bill patients according to this scheme, many do not. The bulk of the research undertaken in this field focuses on the barriers the individuals experiencing homelessness face as clients, whilst little research [49,50,51,52,53,54,55,56] focuses on the service providers, notably their perception of the barriers that their clients face as well as the challenges that the providers themselves encounter when trying to offer care.

Although many studies have suggested suitable approaches to health care, such as outreach services [17,21,29] and nurse-led clinics [29,49,51], to address the health care needs of people experiencing homelessness in an urban context [57,58,59], few studies [49] have recommended approaches to health care specific to the regional context. For the purpose of this study, regional is defined as an area that is in or within 20 km drive of a town with over 50,000 residents [60]. Although Australia has an increasingly urbanized homelessness profile [61], there are no data which specifically compare the rates of homelessness in regional versus urban areas in Australia. Data related to the utilization of specialist homelessness services, however, indicate that since 2012, the largest growth in service demand has been experienced in regional locations [13]. Of note, the minimal research which has been undertaken in a regional/remote location within Australia has focused primarily on the housing issues [62,63,64] and was less concerned with the health needs of this population. Clearly, homelessness is not confined to the cities. Although individuals experiencing homelessness living in regional areas face many of the same challenges confronting individuals experiencing homelessness in urban areas, their experience is distinctive due to higher unemployment rates, limited educational and employment opportunities, transportation challenges, reduced rental availabilities, lack of specialized services, lower incomes, as well as lower housing standards [61,65,66]. This highlights the importance of understanding the potentially specific issues of homelessness in a regional context.

Tasmania presents a unique context in that, despite having the lowest rate of homelessness in Australia at 31.8 people per 100,000 [67], compared with the national rate of 50 people per 100,000 [68], there are some unique social and health issues that place individuals experiencing homelessness in arguably a more precarious position. Such issues include having the largest ageing population nationally, the highest rates of chronic conditions and obesity [69], limited public transportation options [70], the least rental affordability of regional areas compared to other parts in Australia [71], the highest rate of poverty, the lowest level of literacy and numeracy skills, and the lowest levels of education across this population [72]. Another key consideration is that when compared to the national average, Tasmania has higher rates of rough sleeping (10%, compared with the national rate of 6%) and individuals couch surfing (32%, compared with the national rate of 17%) [73]. For the purpose of this study, *rough sleeping* refers to an individual who is forced by circumstances to live on the street or in their car [74]. Individuals sleeping rough or couch surfing often fall below the radar because of the relative invisibility of their situation. This means that their healthcare needs are often overlooked. Due to all of the above factors contributing to this unique situation, it is necessary to undertake a local assessment to augment service provision to ensure that local services are addressing the healthcare needs of this marginalized population. A focus on service providers rather than individuals experiencing homelessness and their families provides an important alternative source of new information to help improve services in line with clients’ often complex needs.

This study therefore seeks to examine the physical and mental health needs of individuals and families experiencing homelessness based in Launceston, Tasmania, from the perspective of service providers.

## 2. Methods

A qualitative descriptive methodology was employed, as this is well suited to providing insight into the experiences and perceptions of the participants [75]. This is of particular importance when the focus is on a poorly understood phenomenon and/or on a topic that has not been researched extensively [76]. Moreover, such a methodology is particularly suited to healthcare contexts, as it employs a naturalistic perspective, allows for flexibility in study design, and uses data collection involving interviews and purposeful sampling [77]. In order to obtain rich narrative data from this study group, participants were engaged through semi-structured interviews [78]; as shown in Table 1, these questions were developed from the literature review. The interviews sought to answer the research questions: What are the predominant health issues of Launceston’s homeless population? What barriers do individuals experiencing homelessness in regional contexts face when accessing healthcare? What challenges do service providers experience when providing services to this population? What approaches to health care would they consider are best suited to effectively address the health issues of Launceston’s homeless population?

### 2.1. Participants

The study participants were service providers working with individuals who were homeless or at high risk of becoming homeless in Launceston, Tasmania. To reflect a diversity of viewpoints, convenience and purposeful sampling methods were used [79]. Such sampling techniques allow for recruitment of a smaller number of participants if the participants are familiar with the research subject [80], as was the case in this study. The targeted population was invited to take part in the study through personal invitation via email. The participants email addresses were publicly available from the service providers’ websites. Only individuals that were (i) 18 years or older; (ii) employees of organisations located in Launceston which were at the time providing services for homeless populations; (iii) voluntary participants; and (iv) able to give informed consent were included in the study. A sample size of 11 was obtained. During the final interviews, few new themes emerged, indicating that data saturation was attained.

### 2.2. Procedure

Ethics approval was obtained from the University of Tasmania’s Human Research Ethics Committee (approval number H0018293). All 11 participants provided written consent after reviewing the Participant Study Information Sheet. Five participants were interviewed one-on-one, whilst six took part in two three-member focus groups. The interviews and focus groups [81] were either held face to face at the participants place of work, or via telephone. The interview guide (Table 1) was developed after a comprehensive review of the literature. The interviews were voice recorded, de-identified and transcribed verbatim by the individual. The interviews took between 20 and 30 min each, and the focus groups took between 20–50 min. Recruitment and interviews continued until there were no new emergent themes or codes [82,83].

### 2.3. Data Analysis: Coding and Thematic Analysis

The transcripts were coded with the aid of NVivo Pro (version 12). Thematic analysis was inductive [84] and followed Braun and Clarke’s [85] seven stages: transcription; familiarization; initial coding; identification of initial themes; reviewing the themes; defining and naming themes; final analysis and reporting of results [85]. Upon careful reading of each manuscript, one member of the research team (GBD) undertook open coding with advice from the other two team members. Open coding meant that pre-set codes were not used [85,86]; instead, common words, synonyms, phrases and ideas were identified, as well as noting trends in data and highlighting emergent themes [87]. A draft of coding provided by the lead author was checked by the other two team members, and feedback was given. Following discussions with the wider research team, the transcripts were coded on three separate occasions by the same investigator. To ensure rigor and trustworthiness, the codes and categories were revised after each coding attempt until they were finalised.

## 3. Results

The participant characteristics including the professional specificities of each participant are shown in Table 2. Four main themes and nine sub-themes emerged from the thematic analysis. These results are presented under the subheadings ‘health issues’, ‘barriers to meeting the healthcare needs of individuals experiencing homelessness’, ‘type of healthcare facility most used/referred to’ and ‘approaches to health care’. These are outlined in Figure 1.

### 3.1. Health Issues

All the participants explained that the poor health status of their clients was an important issue that compounded their already difficult existence. The participants discussed a range of mental and physical health conditions common to their clients.

#### 3.1.1. Mental Health Conditions

All but one participant identified that mental health was a key health issue for their homeless clients, with five participants citing depression and drug and alcohol dependence as prevalent conditions. Anxiety, borderline personality disorder and self-harm, highlighted by four participants, were the second most frequently reported conditions. Four participants also discussed the consequence of poorly managed mental health conditions, explaining that self-harm and suicide was common. One participant explained:


*“Particularly from a perspective around suicide, a multitude of social factors which lead to an inability to cope, and then feelings of disconnection which comes from those experiences”.*
(P2)

#### 3.1.2. Physical Health Conditions

Physical health conditions were also raised as a significant issue by a number of participants. Five of the participants noted endocrine disorders, such as diabetes, as a common health condition for their clients. Three of the participants explained that their client’s diabetes was often poorly managed due to limited access to adequate and nutritious food. The other commonly cited physical health issue was oral health, with four of the participants explaining that this was often a condition that their clients needed help with. Hypertension, emphysema, cancer, sexual health and physical disabilities were also raised as problems for individuals experiencing homelessness.

Bi- and tri-morbidity was also highlighted by four of the participants as a significant problem that impeded their client’s ability to access required health care. One participant explained:


*“Especially if there’s dual diagnosis. If they’re [involved with both] drug and alcohol use. They fall through the gaps because neither side wants to pick them up”.*
(P11)

The high rates of physical and/or mental health conditions reported amongst this population should indicate that this cohort overutilises healthcare services, when the opposite is often true. All participants identified that this population more often underutilised healthcare services due to facing a multitude of barriers, as discussed below.

### 3.2. Barriers to Meeting the Healthcare Needs of Individuals Experiencing Homelessness

This over-arching theme, barriers to healthcare access, offers crucial insight into the barriers that individuals experiencing homelessness face when trying to access healthcare, as well as the barriers that service providers face when trying to provide support to individuals experiencing homelessness. This section has been further divided into three different but often interrelated sub-themes: client-level barriers, provider-level barriers and system-level barriers. The identified barriers for each sub-theme are summarized in Figure 2.

#### 3.2.1. Client-Level Barriers

Of the three sub-themes, client-level barriers were discussed most frequently and in most detail by the participants. The barriers that were cited included: living day-by-day, financial constraints, health literacy, mental health issues, behaviour and safety, as well as stigma.

##### Living Day-by-Day

A significant client-level barrier which repeatedly emerged was ‘living day-by-day’. This barrier indicated that individuals experiencing homelessness often do not prioritise their health, as they have more pressing issues to worry about, such as food, security and shelter. Due to the precariousness of their existence, this often meant that they were unable to commit to attending future health-related appointments, as one participant explained:


*“If a receptionist says there is an appointment in a weeks’ time....it is often missed or completely forgotten about, cause it’s just too hard, they’re not thinking that far ahead, they’re just thinking about getting through that day really”.*
(P8)

##### Financial

Closely related to ‘living day-by-day’, was ‘financial constraints’; nine participants identified this as a significant barrier. This barrier not only pertains to not having the financial means to pay to see a general practitioner (GP) or engage with other community led services, but also to their ability to cover the costs of transportation, as well as having phone credit to arrange an appointment. The financial inability to call to cancel the appointment led to individuals experiencing homelessness being perceived as ‘unreliable’, and they were then often denied future access to that particular service.

##### Health Literacy

Health literacy was repeatedly discussed as a significant barrier to health care access. This was explained in relation to not having the capacity to organise an appointment, not understanding the medical terminology and thus not being able to consent to certain medical procedures, and not being able to follow the doctor’s instructions related to medication regimes or the requirements for follow-up care. This was explained by one participant:


*“The language barrier… [the use of big words and all that. They [the individual experiencing homelessness] just sit there and go ‘yeah, yeah, yeah’. And then I walk out with them and say, ‘did you understand?’ and they say ‘no, not really’. It’s just too hard”.*
(P9)

The five participants that discussed this explained that a number of their clients were unable to navigate the healthcare system effectively, resulting in inappropriate presentations to certain healthcare facilities and/or not following through with obtaining their healthcare needs met.

The health literacy of the client was also identified as something that also affected the service provider in the delivery of their care. The provider highlighted the fact that it was difficult to keep the clients connected with care, as some lacked insight or an understanding of the importance of attending certain appointments that had been organized for them. This often meant that the client never progressed or could be discharged from their services, as their issues were never resolved. Although directly related to the capacity of the client, this was also discussed in relation to the capabilities and capacity of the service providers to meet the needs of their clients. Four participants expressed their concern with their inability to provide the clients with comprehensive support for a number of possible reasons: they did not have the skills; they may not have a thorough comprehension of how to navigate the healthcare system themselves; and/or they did not have the required resources or time to do so.

##### Mental Health Conditions

Having a diagnosed or undiagnosed mental health condition, such as depression, schizophrenia, PTSD and anxiety, was also raised. Poor mental health not only inhibited the individual’s ability to secure and retain somewhere to live, but it also impinged on their ability to follow complex referral processes, to trust the medical professionals, and simply to persist in following self-care instructions. These issues were identified as being even worse when the individuals were not taking their prescribed medications, as explained by one participant:


*“We do have some that … been banned from some of those places [health care providers]. And that can be [due to] mental health problems, been too long without meds... so that can cause some issues”.*
(P9)

Participants stated that they were not equipped with the right skills or training to deal with clients when they were experiencing significant mental health issues. This lack of skills often meant that the participants opted to send clients to the emergency department (ED) rather than other services, as the ED offered a more immediate service.

##### Behaviour and Safety

The behaviour of the homeless clients was raised on three occasions. One participant explained that there were times when they had banned clients from their service for a period of time due to disruptive or aggressive behaviour. Two participants also explained that some of their clients had also been banned from other organisations due to erratic behaviour, which then further limited the choices of where the service provider could then send the client. This then resulted in the client not progressing and their medical situation not improving. One participant explained:


*“Unfortunately, some have developed anger issues …. There are a lot of doctors that they have been known to say “well, you’re not coming back here anymore”… Some of them [the individuals experiencing homelessness] have gotten to a point where they just don’t know where to go anymore”.*
(P8)

##### Stigma

The final client-level barrier identified was stigma. Even when a homeless individual could find a service from which they can receive care, service providers reported that the individuals experiencing homelessness felt stigmatized by their situation. This was discussed in relation to not wanting to seek help, as they did not wish to be identified as homeless and/or did not want to disclose their status, self-consciousness due to needing care, encountering other people experiencing homelessness, or encountering people they knew before they became homeless. All of these reasons have meant that the homeless clients were then less inclined to engage with health care services.

#### 3.2.2. Provider-Level Barriers

Participants also identified several provider-level barriers that affected the ability of individuals experiencing homelessness in obtaining their healthcare needs, including ‘unavailability of a suitable GP’, ‘fragmented services’, ‘limited resources/skills’ and ‘previous bad experiences’.

##### Lack of Availability of a Suitable Doctor

The inability to find a GP that bulk billed (provided free care) was seen as a critical barrier, specific to Launceston. Most participants explained that they were unaware of a single GP that bulk billed that was willing to take on new clients. Closely intertwined with this was the availability of appointments. Those who identified this as a barrier spoke of this in relation to GP clinics, with one also discussing this problem in relation to a specific mental health organization. Some participants reported that clients were often waiting up to three weeks for an appointment. These long wait times often resulted in their clients opting to go to the local emergency department to get basic healthcare needs met or simply not getting their healthcare needs addressed.

Being able to link homeless clients with a GP was also a challenge for the service providers. Six of the participants highlighted this as a significant challenge, as it often meant that they then had to take the homeless client to the ED for basic healthcare. This then required the service provider to wait with the client in ED, often resulting in short staffing at their service. When the service provider would/could not escort the client to ED, the homeless individual often did not follow through with getting their health care needs met.

##### Fragmented Services

Fragmented services pertained to issues in the transfer of healthcare information, as well as complex referral processes. One participant explained:


*“To get into certain programs you need to get referred through this department, but to get into that department you have to be referred to another department…..it’s not quite as streamlined and not as easy to get people into the programs they need to be in”.*
(P1)

Four of the participants expressed the difficulties in providing care to their clients due to fragmented services. This pertained to the challenges of being the ‘middleman’ trying to coordinate care between the client and the healthcare provider, as well as the difficulties in providing the required support when the received referrals contained minimal information and no clear management plan. One client also expanded on this and explained that one of the key difficulties was that the health department and the homelessness sector worked in ‘silos’.

##### Limited Resources/Capacity

Lack of resources repeatedly emerged as a challenge for the service providers. Difficulties associated with high volume of clients, coupled with the limited capacity of services to provide consultation times long enough, often meant that the participants felt as though they were unable to provide the homeless clients with the care and support that they often required. This inability to provide adequate care often resulted in clients no longer using their services. One participant explained:


*“There’s just not that capacity to sit down with someone for 3, 4 h and talk through stuff, sometimes it’s just a 20-min appointment”.*
(P1)

##### Previous Bad Experiences

Previous bad experience with healthcare services was another frequently reported barrier related to the provider. Six participants explained that their clients had reported being treated poorly by certain GP practices as well as ED staff. Four participants explained that concerns are often dismissed, the homeless do not feel safe, and they do not trust the medical professionals. One participant suspected that certain GP clinics discriminated against individuals experiencing homelessness due to the fact that they frequently required longer appointments because of their complex healthcare requirements.

#### 3.2.3. System-Level Barriers

The final sub-theme relates to system-level barriers, which included ‘over-stretched healthcare services’, ‘transportation to appointments’ and ‘funding’.

##### Over-Stretched Healthcare Services

The issue of over-stretched healthcare services was raised on a number of occasions. This was discussed in relation to the limited services available in Launceston. Some simply highlighted a lack of Medicare-covered (free) healthcare, availability of dentists, availability of trained mental health specialists, as well as long wait times at ED.

The availability of adequate mental health support services was something specifically identified as lacking in Launceston. Headspace (the nationally funded youth mental health organisation) was the preferred service for six of the participants, for when a client aged 12–25 years required mental health support. However, there were issues in relation to long wait times for gaining an appointment, which then resulted in the participant needing to take their clients to the emergency department to obtain their mental health needs.

Long wait times in the ED were raised as an issue by four of the participants. Most of these participants explained that they had heard of individuals waiting between four and six hours to simply get a script filled. One participant explained that if their clients were required to wait too long, they would often leave before getting care, resulting in further deterioration of their medical condition, as well as developing additional comorbidities.

Despite the long wait time, the ED was identified by five of the participants as a facility that they frequently sent their clients to. One participant explained:


*“The other choice, because it is the only choice, is the emergency department … they can’t get their needs met elsewhere. I heard … of someone who … needed to get a repeat prescription, but because they hadn’t been for a little bit to the doctor, they had to make an appointment to re-visit … before the script could be written … couldn’t get an appointment, couldn’t get in … ended up presenting at the emergency department to try and get that need met … So, the choice is the emergency department, because there is no other choice for some people to have their needs met”.*
(P6)

This quote illustrates the overlapping nature of barriers to care: access to GP’s that bulk bill, availability of appointments, as well as the overstretched healthcare services that individuals experiencing homelessness approach.

Due to the limited options, participants explained that it often meant that individuals experiencing homelessness were required to travel great distances to obtain their healthcare needs, which leads to the next barrier—transportation.

##### Transportation

Multiple service providers reported lack of appropriate public transportation to appointments as a significant barrier, with walking to healthcare facilities being the principal means of transportation. One service provider explained that for one of their clients to have their healthcare needs met:


*“He needs to scoot (motorised wheelchair) all the way out … and all the way back (12 kms each way). He has to charge it when he’s out there, so he can get back. That’s a long way to go … let alone the safety aspects, I see him on the road sometimes … that’s not a good thing to have to do all that”.*
(P5)

##### Funding

Funding was also raised as a significant constraint. Some participants explained that professional development of staff to better meet client needs was often not prioritised due to limited available funds. This was spoken about in relation to mental health training specifically. Issues related to funding drying up was also discussed in regard to previous approaches to health care even when these were deemed successful. Programs that had previously been implemented were no longer running, as the funding had either been exhausted and/or was re-allocated to different programs.

### 3.3. Approaches to Health Care

The participants were asked about successful approaches to health care provision that they are currently using, have previously used or wished they were using. The provision of outreach services was consistently reported by the participants as their preferred method of care, with seven reporting that they had previously used such a model or had wished such a model would be implemented at their place of work. Many participants cited that they believed their clients would be more likely to engage in healthcare services if these were offered onsite as part of an organisation they were already linked with.

One participant stated that they had an on-site nurse practitioner who provided free healthcare to their clients one morning per week. This participant explained that since the free clinic was established, many clients had expressed their gratitude that the service exists and that they felt better equipped to manage their healthcare needs. This same participant said that although this was a much-needed service, there were still gaps in the provision of care, stating that to provide their clients with holistic care, they would ideally like to expand the service and provide free dental care, podiatry care, mental health support and paediatric specific care. One participant who had previously had an outreach service available for his clients explained:


*“In the early days, we used to have a doctor that would come over once a week to see any residents that need assistance, the doctor retired, and the service stopped. In today’s climate to have a nurse or a doctor available to visit on a weekly basis would have so many benefits to our service and take the pressure of other services”.*
(P8)

Expanding on this was the idea from two participants that the outreach service needed to be linked to the hospital so as to ensure continuity of care and back-up in case emergency interventions were needed.

One participant who managed a long-term housing facility for homeless men had a long-standing memorandum of understanding (MOU) with a local GP clinic. The MOU allowed him to always be guaranteed an appointment and that the client be bulk billed, regardless of whether or not the client had attended the medical centre before. The participant explained that this approach to health care worked exceptionally well. He also explained that they had previously had an agreement with a GP who would visit the facility on a weekly basis to address any simple healthcare needs. The participant expressed that in an ideal world, he would like to provide both services to his clients: an MOU with the GP clinic and to have outreach services made available.

Case management was also discussed as a positive approach to health care and one that should be reinvigorated by certain service providers. To illustrate the complexities of factors that need to be in place for the effective support of this population, one participant explained:


*“It was a case management model. You had social workers, but you had community workers as well, TAFE trained. That would make themselves very available to clients, booking appointments, taking them to appointments, making sure their needs were met, liaising with services, building a team. And we don’t have a service that does that anymore …. I think that we really need to turn it on its head and go out to the patient and make ourselves more accessible”.*
(P2)

The same participant explained that their delivery of care should be underpinned by trauma-informed care principles. Although there has been additional trauma-informed care specific training provided to many healthcare staff in Launceston, the participant explained that due to competing demands and the high volume of patients, it was difficult to fit into an already busy schedule.

Another novel idea was to have a dedicated mental health emergency department, whereby the clients are triaged by, cared for and followed up by mental health specialists. The participant explained that they believed such a service would be not only beneficial for the homeless clients but would reduce the burden on the local emergency department and appeal to the staff in ED who would prefer to deal with physical health conditions.

## 4. Discussion

This study examined the health needs of regionally based individuals experiencing homelessness from the perspective of service providers. Consistent with other research previously undertaken in the United States of America [19,25], United Kingdom [20,21], Canada [35], Europe [40,58], as well as other parts of urban Australia [17], the individuals experiencing homelessness in this study also demonstrate high rates of comorbid physical and mental health conditions. The results confirm the bidirectionality of homelessness and poor mental and physical health, and this greatly impedes their ability to make progress in any of the areas of their needs, whether in housing, education or employment [10,17,22,88]. The interdependencies between all these issues compound and feed into the cyclical nature of homelessness. The results also suggest that the health of regionally based individuals experiencing homelessness compared with those living in urban areas is exacerbated due to regional and Tasmanian specific factors, including the ageing population, high rates of chronic conditions, poor health literacy, lack of affordable housing, limited public transportation options and high rates of poverty [69,70,71,72].

Despite individuals experiencing homelessness having higher rates of all measures of health inequality [16,26,27], consistent with the previous literature [24,38,42], the service providers from this study reported that these individuals experiencing homelessness were not linked with primary and preventative services, and the care that was accessed was inconsistent. The results have exposed a number of client-level, provider-level and system-level barriers to accessing good care. Although most of the identified barriers in this study are similar to studies that have been undertaken in an urban context [39,42,43,44,57,89,90,91,92,93], three identified barriers were uniquely regional: the scarcity of GPs who bulk-bill (provide free services), the impact of lack of transport, and low literacy rates.

First, the most frequently identified provider-level barrier was the lack of availability of GPs that routinely bulk bill. While previous studies have identified that there was a scarcity of GP appointments available [91,92] and that homeless clients required longer appointment times [29], a lack of bulk billing GPs was not identified as a key barrier for their Australian participants and is therefore a unique finding from this study. The scarcity of available GPs willing to bulk bill has become an urgent issue that is particularly apparent in regional areas [94]. This is further compounded by the overall low GP-to-person ratio in Tasmania [95].

The inability to find a suitable GP with available appointments and who offers bulk billing services often means that individuals are required to travel longer distances to obtain their health care needs, which leads to the next barrier: transportation. This of course has particular significance in a regional context. Although identified in studies undertaken in Europe [96], Canada [45] and America [43,45,88,90,91], this barrier was discussed primarily in relation to the financial burden of paying for transportation, with few studies [43,44,45] specifically investigating the lack of suitable public transportation options, which was raised as a relevant barrier in this study. Moreover, only one study [45] was undertaken in a regional location. Regional travel is overwhelmingly car dependent, with poor or non-existent public transportation options available. The combination of these two barriers meant that many individuals living in Launceston opted to go to the local emergency department to obtain their health care needs, inevitably leading to overstretched healthcare services. The emergency department at Launceston General Hospital is often overstretched, and waiting times are long [97]. Presenting to hospital emergency departments when in need of health care is a consistent finding in the broader literature [93].

Another barrier of particular significance in Launceston and other regional locations within Australia was related to literacy. In Tasmania, 49% of the population are classified as illiterate [98]. Literacy is one of the strongest predictors of health status [99,100], with low individual health literacy being associated with higher rates of hospitalisation and emergency care utilization [101]. Of key consideration is that reported lower levels of health literacy are associated with not being housed and having a psychotic mental health condition [46]. Health literacy does not only refer to the individual’s capacity to understand medical advice and procedures but also extends to the health literacy environment. The healthcare providers within the environment relevant to the individual have a responsibility to improve access and equity [102,103].

Although all of these issues are also relevant to the general population, the extent to which these barriers affect individuals experiencing homelessness is exponential, with limited social or financial means to cushion against the impacts of these barriers [45,92]. Addressing these issues for individuals experiencing homelessness and finding enablers and facilitators could clearly improve the opportunities for people experiencing homelessness to manage their health conditions, reduce the likelihood of developing additional health issues and thus in turn improve their ability of securing and maintaining suitable employment as well as stable accommodation.

### 4.1. Potential Approaches to Health Care

Although participants were not specifically asked about approaches to health care, when asked about service improvements, their responses offered insight into potential approaches to health care that could suit the Launceston context. In alignment with this study, previous research has suggested the establishment of free access to primary care workers via outreach services [17,21,29,57,104] as an approach to health care suited to this population. Such a model would not only reduce the burden on local emergency departments but is likely to enable the most appropriate delivery of care [28]. Similar to a study by Roche et al. [29], the benefit of having a healthcare service on site at a homeless support service provider was echoed in this current study. However, unique to this study, the service provided was solely nurse led.

Similar to a study by Keogh et al. [21], the establishment of an MOU with a local GP was seen as particularly effective for the service provider as well as for the clients who are homeless, as it ensured that they had access to a GP who was guaranteed to bulk bill. However, due to the lack of availability of suitable GPs, the establishment of free nurse-led clinics as the first point of contact may be more suited to the Launceston context. Such clinics are not only well placed to improve access to healthcare and enhance the continuity of care for individuals experiencing homelessness [46], they also reduce the burden on local emergency departments [50,51]. Additional research is required to explore if nurse-led clinics provide a suitable alternative to the more traditional provisions of care for individuals experiencing homelessness.

Findings from this study highlight that the current approach to addressing the healthcare needs of individuals experiencing homelessness needs to be re-evaluated. Although addressing the larger issue of accessible and appropriate housing is crucial, there are specific measures that need to be implemented to address the health inequalities. Mental health has repeatedly been reported as an issue for this population. This emphasises the importance of ensuring the establishment of partnerships between the existing mental health services and the current homeless support providers. Such collaborations would ensure that the long-term mental health care support is provided to individuals experiencing homelessness in a manner sensitive to their needs and the specific challenges of their lives. Overall, access to GPs who are not only willing to offer bulk billing services but also longer consultation times will be key. If there is an inability to attract more GPs to provide bulk billing services, then the establishment of more nurse-led clinics would be necessary [29,42,50,51]. Regardless of which approach to health care is deemed to best engage individuals experiencing homelessness in healthcare, there is a need to ensure that the barriers which people experiencing homelessness face are eliminated or at least reduced.

### 4.2. Limitations

A number of limitations have been identified within the study. The study captured the perspectives of services providers that were employed as housing support workers, social workers, nurse practitioners and case workers. Mental health specialists and general practitioners were invited to take part in the study but declined. Another limitation is that the data collected were only from service providers. The perceived needs of service providers may not be entirely congruent with the needs prioritized by the individuals who are homeless who are accessing or not accessing services; this is of particular importance in relation to the client-level barriers identified. Furthermore, as the study was undertaken in Launceston to understand the local context, the results may not be transferable to other regional areas in Australia. Future studies should address these aspects to further understand the unique impact of homelessness in the regional Australian context.

## 5. Conclusions

The findings from this study highlight the complex health needs of individuals experiencing homelessness specific to the regional context of Launceston, Tasmania. While previous studies on homeless individual’s health issues have largely focused on the perspective of the homeless individual, this study has addressed a gap in the research by investigating the challenges that service providers face when providing support services to individuals experiencing homelessness. The findings suggest that while this current study offers a new dimension to the research, further research is needed to more fully understand the experience and situation of individuals experiencing homelessness in Launceston. The next step should be a thorough investigation of the lived experience of people experiencing homelessness to not only gain an understanding of the common health concerns of this population and the barriers that they face when trying to obtain their healthcare needs, but to also to ensure that services can be appropriately augmented to address their health care needs. The data from this study have highlighted that the health needs of individuals experiencing homelessness are multifactorial, and thus in addressing these needs, the strategies must also be multifaceted and flexible. The establishment of free access to primary care workers via outreach services will clearly be an essential starting point.

Addressing the health issues of Launceston’s homeless population is only the beginning. Although not raised in this study, to provide these individuals with a chance of regaining control over their lives so as to be able to secure employment, the housing situation also needs to be addressed [27,39]. Communication with all the other relevant service providers addressing the issues of homelessness is crucial so that a coordinated and maximally effective approach is supported. With all the services pulling together, it is hoped that homelessness in Launceston and other regional centres will be reduced if not eliminated, and those individuals who have suffered homelessness are finally able to be integrated back into the community.

## Figures and Tables

**Figure 1 ijerph-19-08368-f001:**
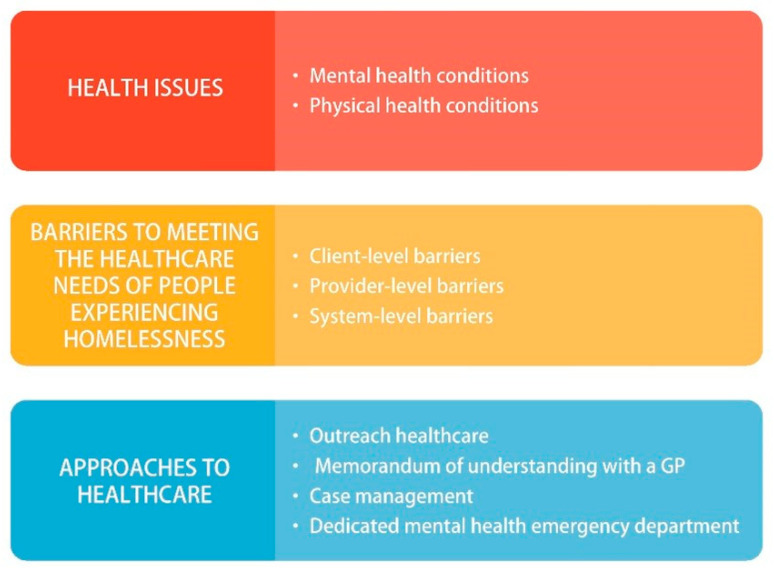
Themes and sub-themes.

**Figure 2 ijerph-19-08368-f002:**
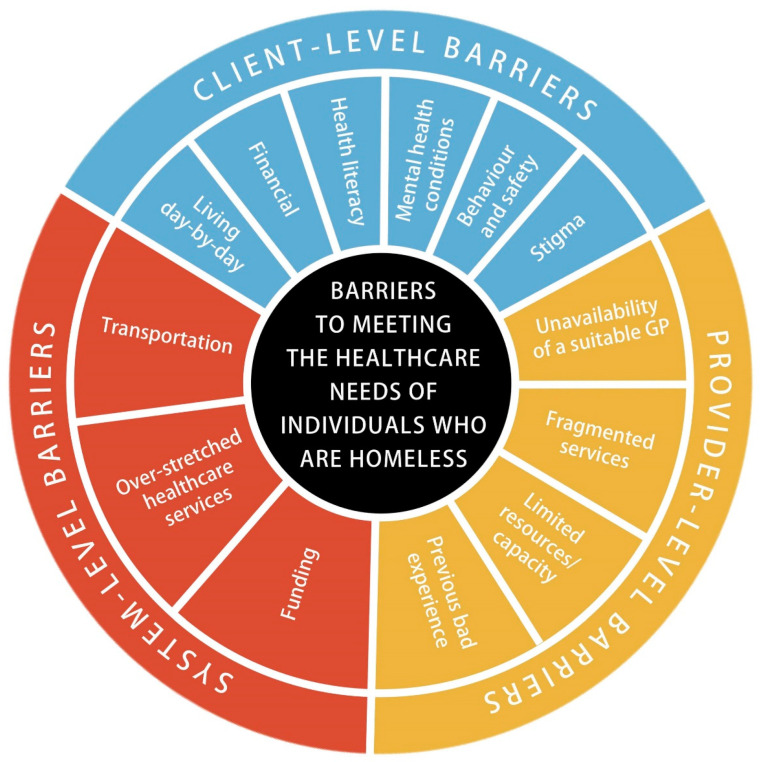
Barriers to meeting the healthcare needs of individuals experiencing homelessness.

**Table 1 ijerph-19-08368-t001:** Interview and focus group interview guide.

Questions
**General**
1. What is your position within the agency? What does your role involve? Which programs are you responsible for? How long have you been in the role?
2. Can you tell me a little bit about the services that your organisation provides?
**Service provision**
1. What in your experience are the main health issues that people experiencing homelessness need services for?
2. Where are people experiencing homelessness currently seeking health support from?
3. How do health and community services address the needs of the homeless? What needs go unmet?
4. What do you think are the barriers and challenges for people experiencing homelessness in accessing healthcare?
5. What barriers do you face in providing services/care for the homeless population?
6. What factors do you think assist the homeless in accessing healthcare?
7. Are there any particular services that you believe need to be provided/improved? How can services be better provided?
8. Does your organisation refer clients onto healthcare services? If so, how does this work and to which organisations? How successful is this referral process?
9. Is there anything else you would like to add?

**Table 2 ijerph-19-08368-t002:** Participant characteristics.

Participant Number	Agency Type	Role of Agency	Position in Agency
1	Community agency	Housing support (crisis accommodation)	Client support worker
2	Hospital	Provision of healthcare	Social worker
3	Hospital	Provision of healthcare	Social worker
4	Community agency	Provision of healthcare	Nurse Practitioner
5	Community agency	Community Centre	General manager
6	Community agency	Emergency relief	Case Manager
7	Community agency	Housing support (long-term accommodation)	General manager
8	Community agency	Emergency relief	Case worker
9	Community agency	Housing support (crisis accommodation)	Client support worker
10	Community agency	Housing support (crisis accommodation)	Client support worker
11	Community agency	Housing support (crisis accommodation)	Client support worker

## Data Availability

These data sources are not publicly available, and ethics approval has not been provided to share data at the time of publication.

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
