# Peer review of "The Health Needs of Regionally Based Individuals Who Experience Homelessness: Perspectives of Service Providers"

_ijerph, 2022, doi:10.3390/ijerph19148368_

Round 1
Reviewer 1 Report
This study seems well thought out and makes a good contribution to existing literature. I suggest a Revise and Resubmit with the following edits:
· Great literature review on prior findings on health with particular attention to the semantics of what is still unclear. Great job.
· Mental illness/health among homeless populations is still very unclear. Findings literally range between 4–100% (Fischer 1989; Lee et al. 2010; Shlay and Rossi 1992). Among the reason for such a large range is due to the various use of methods to study this population (susser et AL 1990).
· What “regional” areas means needs to be established up front. I found it unclear what the details of this word meant the first time I read through the paper. Does it mean “rural?” With this in mind, police usually transport people experiencing homelessness to urban centers because that’s where the services are. Please acknowledge this in the text somewhere.
· When you say “Data related to the utilization of specialist 91 homelessness services, however, indicates that since 2012 the largest growth in service 92 demand has been experienced in regional locations” this is a great example of the consideration to detail that I enjoyed while reading this manuscript. However, I would like to see the authors use this kind of language to specify to readers that the homeless population is far broader than those who seek services. Those who seek services tend to be (1) those who “fit” the demographics of what services seek to provide their aid to (2) what funding is available to provide aid for the services to stay open. All this filters out the demographics of what potential respondents are to be studied (sampled) versus the demographics who experience homelessness but do not (even avoid) services. In short, prior research only knows the demographics (and situations) of those who seek services (i.e. the demographics that funding for services seeks). Those who avoid services are literally ignored bu research. *Please also put something like this in the limitations section too. So many people who experience homelessness need this to be said for policy provision. Otherwise, research will merely reflect what services already target via funding from the Continuum of Care (i.e. “chronic homelessness” with disabilities) and overlook anyone who does not fit the demographics services are looking for.
· The phrase “from the perspective of service providers” keeps begging to be elaborated upon. Helter Shelter by Prashan Ranasinghe captured the perspectives of social workers. Also, Homelessness and Housing Advocacy: the Role of Red-Tape Warriors also captured the perspective of social service workers. Both books should be mentioned if this is what the authors are trying to capture, but so far there is little in the literature review to the concept of the perspective of social service workers. As I read, I’m left thinking the study will just include a few statements from social workers reinforcing their need for funding (i.e. the demographics their funding sources stipulate they need to stay open), as previously mentioned in my comments above.
· Good citing Patton. However, I’d like to know what informed the questions in the interviews. Could these questions have been made before going into the field? Saying “A qualitative descriptive methodology was employed” is pretty vague. Other than that, the Methods look pretty solid.
The manuscript seems solid in many areas, and is needed, specifically because the way it is written is specific to what many researchers on the subject overlook. Homelessness has been studied since the late 1800s in Chicago, and we know quite a bit about the subject. The authors seem to have an eye for some details and limitations on the subject, specific to measuring health among the homeless population. If the authors make the changes I suggest, then this could be a great contribution.
References
Fischer, Pamela J. 1989. Estimating prevalence of alcohol, drug, and mental health problems in the contemporary homeless population: A review of the literature. Contemporary Drug Problems 16: 333–90.
Lee, Barrett, Kimberly Tyler, and James Write. 2010. The new homelessness revisited. The Annual Review of Sociology 36: 501–21.
Shlay, Anne, and Peter Rossi. 1992. Social science research and contemporary studies of homelessness. Annual Review of Sociology 18: 129–60.
Susser, Ezra, Sarah Conover, and Elmer Struening. 1990. Mental illness in the homeless: Problems of epidemiologic method in surveys of the 1980s. Community Mental Health Journal 26: 3

Author Response
Please find response to feedback attached.

Reviewer 2 Report
Please see attached document.

Author Response
Please find the response to the feedback attached.

Round 2
Reviewer 2 Report
Please see attached document. I commented directly on the PDF. The various changes from the prior version are appreciated and acknowledged, though additional changes will drastically improve the quality.

Author Response
Please find documents attached.
Kind regards,
The research team.
